# Uninterrupted or Minimally Interrupted Direct Oral Anticoagulant Therapy is a Safe Alternative to Vitamin K Antagonists in Patients Undergoing Catheter Ablation for Atrial Fibrillation: An Updated Meta-Analysis

**DOI:** 10.3390/jcm9103073

**Published:** 2020-09-24

**Authors:** Máté Ottóffy, Péter Mátrai, Nelli Farkas, Péter Hegyi, László Czopf, Katalin Márta, András Garami, Márta Balaskó, Emőke Pótóné-Oláh, Alexandra Mikó, Ildikó Rostás, Bastian Wobbe, Tamás Habon

**Affiliations:** 1First Department of Medicine, Division of Cardiology, Medical School, University of Pécs, 7622 Pécs, Hungary; ottoffy.m@gmail.com (M.O.); laszlo.czopf@aok.pte.hu (L.C.); bastianwobbe@gmx.de (B.W.); 2Institute for Translational Medicine, Medical School, University of Pécs, 7622 Pécs, Hungary; matrai.peti@gmail.com (P.M.); farkas.nelli@gmail.com (N.F.); hegyi.peter@pte.hu (P.H.); katalin.martak@gmail.com (K.M.); andras.garami@aok.pte.hu (A.G.); marta.balasko@aok.pte.hu (M.B.); potoneoe@gmail.com (E.P.-O.); miko.alexandra@pte.hu (A.M.); ildiko.rostas@aok.pte.hu (I.R.); 3János Szentágothai Research Center, University of Pécs, 7622 Pécs, Hungary

**Keywords:** anticoagulation, atrial fibrillation, catheter ablation, stroke, bleeding, meta-analysis

## Abstract

Adequate anticoagulation during catheter ablation (CA) for atrial fibrillation (AF) is crucial for the prevention of both thromboembolic events and life-threatening bleeding. The purpose of this updated meta-analysis is to compare the safety and efficacy of uninterrupted and minimally interrupted periprocedural direct oral anticoagulant (DOAC) protocols and uninterrupted vitamin K antagonist (VKA) therapy in patients undergoing CA for AF based on the latest evidence. Randomized controlled trials, prospective observational studies, and retrospective registries comparing DOACs to VKAs were identified in multiple databases (Embase, MEDLINE via PubMed, CENTRAL, and Scopus). The primary outcomes were stroke or transient ischemic attack (TIA), major bleeding, and net clinical benefit. Forty-two studies with a total of 22,715 patients were included in the final analysis. The occurrence of major bleeding was significantly lower in patients assigned to uninterrupted DOAC treatment compared to VKAs (pooled odds ratio (POR): 0.71, confidence interval (CI): 0.51–0.99). The pooled analysis of both uninterrupted and minimally interrupted DOAC groups also showed significant reduction in major bleeding events (POR: 0.70, CI: 0.53–0.93). The incidence of thromboembolic events was low, with no significant difference between groups. This updated meta-analysis showed that DOAC therapy is as effective as VKA in preventing stroke and TIA. Minimally interrupted DOAC therapy is a non-inferior periprocedural anticoagulation strategy; however, uninterrupted DOAC therapy showed superiority compared to VKA with regard to major, life-threatening bleeding. Based on our in-depth analysis, we conclude that both DOAC strategies are equally safe and preferable alternatives to VKAs in patients undergoing CA for AF.

## 1. Introduction

Atrial fibrillation (AF) is the most common sustained arrhythmia in adults, and its prevalence is expected to increase notably during the next three decades [1]. AF is associated with a nearly five-fold increase in the risk of stroke compared to the general population [2]. Oral anticoagulation significantly reduces the risk of ischemic stroke and mortality in AF patients [3]: in a well-managed anticoagulated AF population, the annual incidence and mortality of stroke are around 1.5% and 3%, respectively [4,5,6,7].

Rate and rhythm control may improve symptoms of AF and preserve cardiac function; however, previous studies failed to show a clear benefit in terms of long-term morbidity or mortality [8,9]. Therapeutic strategies for rate or rhythm control have substantially improved over the past few decades, with catheter ablation (CA) gaining increasing importance. The 2020 guidelines of the European Society of Cardiology (ESC) recommend CA for pulmonary vein isolation (PVI) for rhythm control after failure of drug therapy to improve symptoms of AF recurrences in patients with paroxysmal or persistent AF without major risk factors for AF recurrence (Class I A) [10].

Previous reports have shown that about 5–7% of patients undergoing AF ablation experience severe periprocedural complications, and two to three percent experience life-threatening, but usually manageable complications [10,11,12]. The most severe complications include stroke and transient ischemic attack (TIA) (<1%), cardiac tamponade (1–2%), pulmonary vein stenosis, and severe esophageal injury. Silent cerebral ischemic lesions (SCIL) (i.e., white matter lesions detectable by brain magnetic resonance imaging MRI) have been observed in 5–20% of patients treated with radiofrequency or cryoballoon ablation [13].

Adequate periprocedural anticoagulation is crucial to prevent both thromboembolic and bleeding complications. Earlier studies favored uninterrupted vitamin K antagonist (VKA) therapy; however, increasing evidence has demonstrated the similar efficacy and safety profile of DOACs compared with VKA in the context of CA [13,14,15,16]. According to the recent ESC guideline, for patients undergoing AF catheter ablation who have been therapeutically anticoagulated with warfarin, dabigatran, rivaroxaban, apixaban, or edoxaban, the performance of the ablation procedure without oral anticoagulant (OAC) interruption is recommended (Class I A) [10]. However, a recent European survey revealed that most centers use a minimally interrupted direct oral anticoagulant (DOAC) strategy to avoid bleeding complications [17].

The heterogeneity of currently applied DOAC strategies in AF patients undergoing catheter ablation emphasizes the need for further data and consensus recommendations. Therefore, we performed a meta-analysis of all currently available evidence including the most recent trials to examine patterns of periprocedural anticoagulation strategies and associated outcomes in a large patient population and performed sensitivity analysis as well.

## 2. Methods

### 2.1. Search Strategy and Data Extraction

This meta-analysis was designed in accordance with the Cochrane Collaboration recommendations and the Preferred Reporting Items for Systematic Reviews and Meta-analysis statement [18] and registered on the PROSPERO database (CRD42017073075).

Two authors (M.O. and P.M.) independently searched Embase, MEDLINE via the PubMed, Cochrane Central Register of Controlled Trials (CENTRAL), and Scopus databases until March 2020. The search was structured using the PICO format, where the target population was patients after CA for AF, the periprocedural DOAC anticoagulation strategy was compared to VKA treatment, and the outcomes were thromboembolic (efficacy) and bleeding (safety) events. The query involved the following terms: dabigatran, rivaroxaban, apixaban, edoxaban, VKA, atrial fibrillation, and catheter ablation. Only articles published in English were included.

### 2.2. Eligibility Criteria

Inclusion criteria were: (1) patients undergoing CA for AF; (2) the use of the same anticoagulant (DOAC or VKA) therapy during the periprocedural period; (3) uninterrupted VKA treatment; (4) the publication of at least one safety or efficacy outcome; and (5) the presence of a control (VKA) group within the same population. Randomized controlled trials (RCTs), prospective observational studies, and retrospective analyses were included in this meta-analysis.

### 2.3. Screening

After an initial screening of titles, abstracts were screened independently, and eligibility was assessed according to the inclusion criteria. Remaining articles were reviewed in full text. Relevant data from articles were imported into an Excel sheet. These included: stroke, TIA, systemic embolization, major and minor bleeding events, and baseline characteristics. Any disagreements or uncertainties were resolved by discussion, if necessary, with the inclusion of a third party (T.H.).

### 2.4. Endpoints

The primary safety outcome was major bleeding which included pericardial effusion (PE) requiring pericardiocentesis or surgery, retroperitoneal hematoma, hemothorax, and any bleeding requiring transfusion, as defined by the Bleeding Academic Research Consortium (BARC) [19] or the International Society on Thrombosis and Hemostasis (ISTH) [20].

Primary efficacy outcomes were stroke and TIA. Stroke was evaluated based on clinical parameters. Silent cerebral ischemic lesions detected by MRI were not analyzed. The secondary outcome was minor bleeding, defined as any bleeding (including small, asymptomatic pericardial effusions) not requiring intervention.

Net clinical benefit was a composite of primary safety and efficacy outcomes.

### 2.5. Subgroup Analyses

The DOAC groups were categorized based on the number of doses held before the ablation procedure. Interrupted DOAC therapy meant that the last dose was administered more than 24 h before the procedure. The therapy was considered uninterrupted when all scheduled doses were taken regardless of the day of the procedure. When only a single dose of dabigatran or apixaban was held prior to the CA procedure, the therapy was considered minimally interrupted. Since current guidelines do not recommend DOAC interruption, interrupted DOAC therapy was not in the focus of this analysis; data regarding these findings can be found in the Appendix A.

In our study, RCTs and non-RCT studies were distinguished and evaluated both separately and collectively.

### 2.6. Statistical Analysis

The statistical analysis was conducted with Stata 11 SE (StataCorp LLC, College Station, TX, USA). The numbers of patients with observed safety and efficacy outcomes in the DOAC and control groups (VKA) were used to calculate odds ratios (ORs). An OR > 1 indicates an elevated risk of outcome in the DOAC group (favors VKA), whereas an OR < 1 indicates a higher risk in the VKA group (favors DOACs). ORs were pooled (POR) using the Peto method as recommended for the analysis of rare events in the Cochrane Handbook. Summary OR estimation, *p* values, and 95% confidence intervals (CI) were calculated and presented in forest plots. *p* < 0.05 was considered as a significant difference from a summary OR of 1.

Statistical heterogeneity was analyzed using the I^2^ statistic and the chi-squared test to ascertain probability values; *p* < 0.1 was defined as indicating significant heterogeneity.

The small-study effect was tested with Egger’s test, with *p* < 0.1 indicating the sign of bias. In the event of a significant Egger’s test, the trim and fill algorithm was used to investigate the effect of potential bias.

## 3. Results

The literature search yielded 851 articles from the four databases. After removing duplicates and screening titles and abstracts, seventy-three articles remained and were assessed in full text for eligibility. Forty-two studies were included in the final analysis, encompassing a total of 22,715 patients (Figure 1) (references to the included studies can be found in the Appendix A).

Of the 42 studies, twenty-three compared uninterrupted DOAC therapy, and fourteen compared minimally interrupted DOAC therapy to uninterrupted VKA treatment. Twenty-four studies were performed with dabigatran, 16 with rivaroxaban, 15 with apixaban, and 3 with edoxaban. Six of the 42 studies were RCTs. Cryoballoon ablation was used in eight studies, with four studies using this technique exclusively or primarily. The difference between baseline characteristics was negligible. The average age was 62, 62.6, and 61.8 years; the percentages of males were 69.1, 75.6, and 73.9 percent; the average BMI was 27.8, 27.3, and 29.8 kg/m^2^ in the uninterrupted, minimally interrupted, and interrupted groups, respectively. Target activated clotting time (ACT) was always above 300 s (Appendix A; detailed baseline characteristics can be found in Appendix A).

### 3.1. Analysis of RCTs

A total of six RCTS with 2575 patients were included in the analysis, of which five studies applied uninterrupted therapy. In one study, dabigatran therapy was minimally interrupted before CA (the timing of discontinuation was based on the scheduled time of ablation and the treating physician’s discretion) [21].

There was no significant difference between DOAC and VKA therapy regarding thromboembolic events (POR: 0.66; 95% CI: 0.17–2.64) (Appendix A). However, DOAC therapy was associated with significantly fewer major bleeding events events (POR: 0.36, 95% CI: 0.21–0.62) (Appendix A). A statistically significant net clinical benefit (a composite of thromboembolic and major bleeding events) was observed when comparing DOAC treatment to VKAs (POR: 0.35; CI: 0.17–0.73) (Appendix A).

### 3.2. Pooled Analysis of Uninterrupted and Minimally Interrupted DOAC Studies

The occurrence of thromboembolic events was rare, regardless of anticoagulation protocol. Neither uninterrupted nor minimally interrupted DOAC therapy showed significant differences when compared to uninterrupted VKA (POR: 0.95; 95% CI: 0.46–1.96 and POR: 0.96; 95% CI: 0.35–2.64, respectively) (Figure 2).

Major bleeding events were significantly reduced in the uninterrupted DOAC group compared to the VKA group (POR: 0.71, 95% CI: 0.51–0.99), while minimally interrupted therapy did not reach a significant difference (POR: 0.79; 95% CI: 0.43–1.48) (Figure 3).

The composite analysis of thromboembolic and major bleeding events did not reveal any significant difference between treatment groups (POR: 0.74, CI 0.54–1.01, and POR: 0.79, CI: 0.40–1.55; for uninterrupted and minimally interrupted, respectively) (Figure 4). However, the pooled analysis of both DOAC regimens combined showed a net clinical benefit compared to VKA, with fewer major bleeding events (POR: 0.73, 95% CI: 0.54–0.98, and POR: 0.70, 95% CI: 0.53–0.93, respectively) (Figure 4 and Figure 5). There were a total of 611 minor bleeding events registered across the studies included in the analysis. A similar event rate was found in either uninterrupted or minimally interrupted DOAC strategies compared to VKAs (POR: 0.94, 95% CI: 0.78–1.15, and POR: 0.99, 95% CI: 0.68–1.43, respectively) (Appendix A).

### 3.3. Analysis of Nterrupted DOAC Studies

Nine studies used only interrupted DOAC strategies in a total of 3970 patients, all of which were observational in design. Interrupted DOAC therapy showed no significant difference in the primary endpoint compared to VKA therapy (POR: 0.79, 95% CI: 0.45–1.39, and POR: 0.97, 95% CI: 0.28–3.32; for major bleeding and thromboembolic events, respectively) (Appendix A). However, the interrupted DOAC strategy was associated with significantly fewer minor bleeding events than VKA (POR: 0.57, 95% CI: 0.37–0.87) (Appendix A).

### 3.4. Quality Assessment

The quality of individual studies was evaluated using the Newcastle–Ottawa Scale for non-randomized studies and the Cochrane risk of bias tool for RCTs (Appendix A). A funnel plot analysis of the composite endpoint showed a symmetrical distribution of the included studies, indicating no evidence of publication bias (Figure 5).

## 4. Discussion

This meta-analysis examined the incidence of thromboembolic and bleeding complications associated with catheter ablation of AF among patients treated with different anticoagulation strategies. There was no statistically significant difference in thromboembolic complications between uninterrupted DOAC, minimally interrupted DOAC, or uninterrupted VKA strategies. However, DOAC therapy was associated with significantly fewer major bleeding events, which was mainly driven by reductions in the uninterrupted DOAC group. Furthermore, we found a statistically significant net clinical benefit of DOAC treatment compared to VKA.

AF is the most common arrhythmia requiring medical attention, which affects millions worldwide. Traditionally, CA has been a frequently used approach for the restoration of sinus rhythm in AF, despite little evidence for an improvement in hard outcomes, such as mortality and stroke [22]. The recently published CABANA study failed to demonstrate the superiority of CA over medical therapy for AF in terms of reducing death, disabling stroke, serious bleeding, or cardiac arrest due to AF; however, CA was associated with enhanced improvements in quality of life [23]. In contrast, data from Swedish health registries showed that CA was associated with a 30% relative reduction in ischemic stroke, especially in patients with CHA_2_DS_2_-VASc scores > 2.2 [24]. The latest evidence for the reduction of major CV events with catheter ablation was recently published from the EAST-AFNET 4 trial. Early rhythm-control therapy was associated with a 21% lower risk of composite cardiovascular outcomes (death from cardiovascular causes, stroke, or hospitalization with worsening of heart failure or acute coronary syndrome) compared to usual care among patients with early AF and CV conditions [25]. Furthermore, in the randomized, controlled CASTLE-AF study, AF ablation resulted in a significant 38% reduction in the composite endpoint of death from any cause or hospitalization for worsening heart failure compared to medical therapy among patients with AF and heart failure [26].

Our study confirmed the low incidence of thromboembolic complications associated with catheter ablation, with no significant differences observed between uninterrupted, minimally interrupted, or interrupted DOAC therapy compared to VKA. In the minimally interrupted and uninterrupted DOAC groups, nine out of 28 studies reported no stroke or TIA at all, despite a relatively high number of patients involved in these trials (*n* = 109–763). In the rest of the studies, a total of 34 stroke events occurred (<0.2%). The incidence of stroke and TIA was particularly low in studies conducted at large centers with experienced staff, which ensured appropriate perioperative anticoagulation management. The composite analysis of stroke/TIA and major bleeding yielded similar results to the analysis of major bleeding alone, translating to a significant net clinical benefit for DOACs over VKA, without any heterogeneity in the group. In line with our current observations, a recent meta-analysis by Ha et al. found that both minimally interrupted and uninterrupted DOAC regimens were safe and non-inferior alternatives to uninterrupted VKA therapy in AF ablation. This pooled analysis also involved the ABRIDGE-J trial, which showed that minimally interrupted dabigatran therapy resulted in less major bleeding events compared with uninterrupted warfarin with similar thromboembolic safety [27].

While a previous meta-analysis by Cardoso et al. found less major bleeding with continuous DOAC therapy versus similar VKA therapy, an analysis—involving only RCTs—conducted by Romero et al. found that uninterrupted DOAC regimens only demonstrated a trend towards fewer major bleeding events, but this did not reach statistical significance over uninterrupted VKA therapy [28,29].

Accordingly, we found a significant reduction in the risk of major bleeding with uninterrupted DOAC treatment compared to uninterrupted VKA and a trend towards a lower risk of major bleeding with the minimally interrupted DOAC strategy vs. VKA. The pooled analysis of both DOAC strategies also showed a significant difference compared to VKA. This overall benefit was driven by the homogeneous safety benefit of uninterrupted administration design applied in most of the RCTs included in the analysis.

Our observations are also in line with previous findings of Nakamura et al., supporting the safety of both minimally interrupted and uninterrupted DOAC anticoagulation strategies [30]. In this study, the uninterrupted and interrupted DOAC groups revealed a similar incidence of SCILs (19.8% vs. 22.0%, *p* = 0.484) and percentage of SCILs with disappearance on follow-up MR imaging (77.8% vs. 82.1%, *p* = 0.428).

No major bleeding was reported in two studies in the minimally interrupted group. The rate of major bleedings in the VKA arm was similar in both groups (2.2%), although there were two important RCTs, both proving the superiority of dabigatran (one in the uninterrupted group and also one in the minimally interrupted group), in which a much higher (6.9% and 5.0%) major bleeding rate was reported for the VKA arm. Several possible reasons explain this increase. Bleeding events in the RE-CIRCUIT trial and in the ABRIDGE-J study were defined using ISTH criteria, which have a lower threshold for hemoglobin loss than the definition used in some other trials [31,32].

Sensitivity analysis showed that omitting the data from the RE-CIRCUIT trial would reduce the difference to non-significant when comparing major bleeding in uninterrupted DOAC studies and also the composite endpoint in RCTs only (Appendix A), showing that the performance of minimally interrupted DOAC strategy is only marginally inferior to uninterrupted dosing.

There is evidence from RCTs for dabigatran at either uninterrupted or minimally interrupted dosing, and both have a superior safety profile compared to VKA therapy. A recent randomized clinical study reported no stroke and similarly low rates of major bleeding between uninterrupted and minimally interrupted apixaban [33]. For other DOACs, similar efficacies and safety profiles were observed, compared to uninterrupted VKA if either an uninterrupted or minimally interrupted protocol was used.

The 2017 HRS/EHRA/ECAS/APHRS/SOLAECE Expert Consensus Statement is equivocal on the periprocedural management of oral anticoagulation in patients undergoing CA for AF, recommending that one or two doses of DOACs be withheld prior to the procedure or no interruption be used [34]. However, the most recent ESC guidelines clearly recommend performing the procedure without OAC interruption (Class I A) [10]. Despite the availability of expert recommendations, real-world anticoagulation patterns in patients with AF undergoing CA are heterogeneous, and physicians often prefer the interruption of DOAC therapy to avoid bleeding complications. The perceived unpredictability of DOAC’s effects may be due to the lack of routine coagulation monitoring, which causes concerns regarding the safety of an uninterrupted regimen. Accordingly, a European survey (ESS-PRAFA) demonstrated that most centers apply a minimally interrupted DOAC strategy to avoid bleeding complications [17]. Our findings provide reassurance regarding the guideline-recommended use of DOAC strategies in AF patients undergoing CA by showing clear safety benefits for both the uninterrupted and minimally interrupted DOAC strategies over VKA, without a compromise in efficacy. The real-world impact of the new ESC guidelines is yet to be seen; however, our study provides valuable information on the safety and efficacy of different anticoagulation strategies peri-ablation and may serve as further guidance for decision-making in routine clinical practice.

## 5. Conclusions

This comprehensive meta-analysis compared the efficacy and safety of uninterrupted or minimally interrupted DOAC therapy and uninterrupted VKA in a large population of patients undergoing catheter ablation for AF. The risk of thromboembolic events was very low during the periprocedural period with well-managed anticoagulation, regardless of the drug of choice. DOAC therapy was non-inferior to VKAs in the prevention of stroke and TIA and showed safety benefits in terms of bleeding complications.

Based on the currently available data and our in-depth meta-analysis, the widely used anticoagulation strategy of DOACs without or with minimal interruption may be the best practice in the perioperative period of CA for AF.

## 6. Limitations

This meta-analysis has some limitations. Two studies included in the analysis were conference abstracts, which provided limited information about procedural data and patient characteristics. There was also some variation in the criteria for major bleeding in different studies. Furthermore, some studies provided no information on minor bleeding. SCILs were scarcely reported, with no sufficient data available for analysis. Most of the analyzed studies were observational in design, thus representing the foremost limitation despite the satisfactory bias analysis. We did not have access to individual patient data; therefore, we were unable to analyze specific patient, therapy, or ablation characteristics that may have affected clinical outcomes. Finally, a network meta-analysis that directly compares different anticoagulation strategies would also be useful to resolve this clinical problem in the near future.

## Figures and Tables

**Figure 1 jcm-09-03073-f001:**
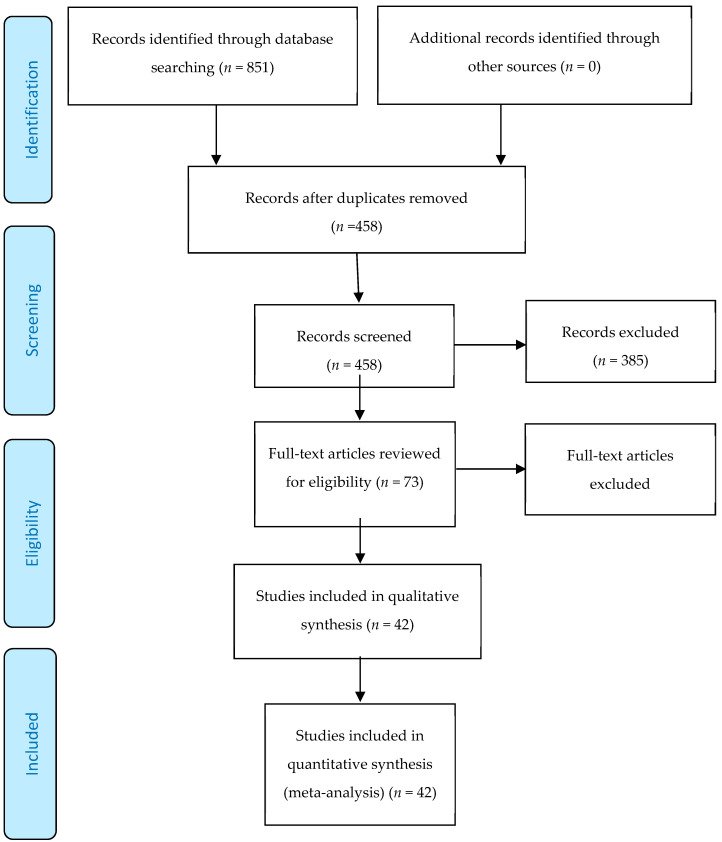
Prisma flowchart of screening and selection.

**Figure 2 jcm-09-03073-f002:**
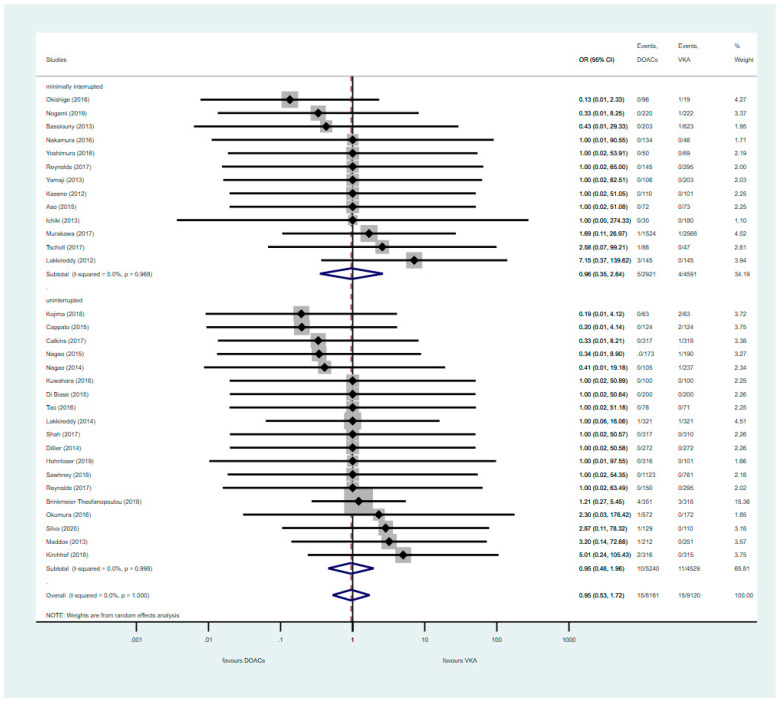
Primary efficacy outcome of stroke or transient ischemic attack (TIA) with uninterrupted or minimally interrupted DOAC strategies vs. VKA. CI: Confidence Interval; OR: Odds Ratio; DOAC: Direct Oral Anticoagulants; VKA: vitamin K antagonist. The diamond defines the overall result of each given group, the dotted red line is there to graphically show the overall results of the groups.

**Figure 3 jcm-09-03073-f003:**
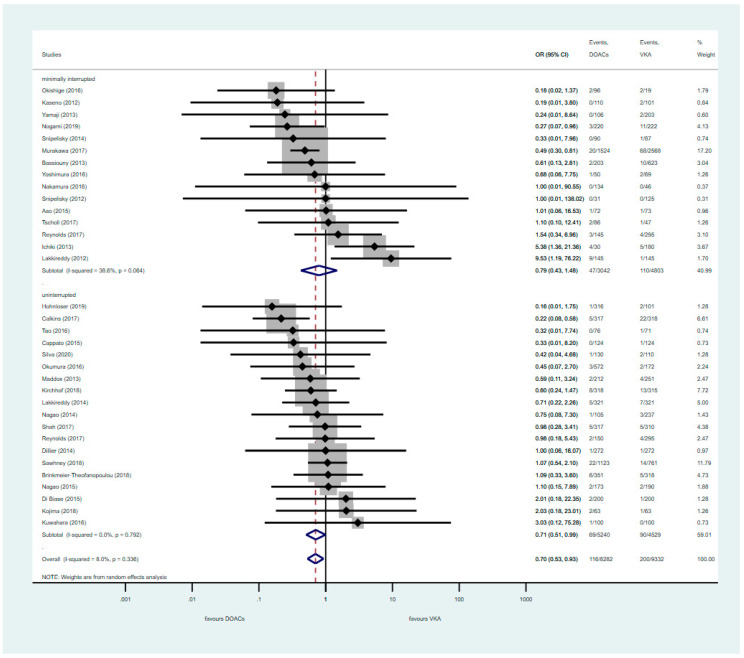
Primary safety outcome of major bleeding with uninterrupted or minimally interrupted DOAC strategies vs. VKA. CI: Confidence Interval; OR: Odds Ratio; DOAC: Direct Oral Anticoagulants; VKA: vitamin K antagonist. The diamond defines the overall result of each given group, the dotted red line is there to graphically show the overall results of the groups.

**Figure 4 jcm-09-03073-f004:**
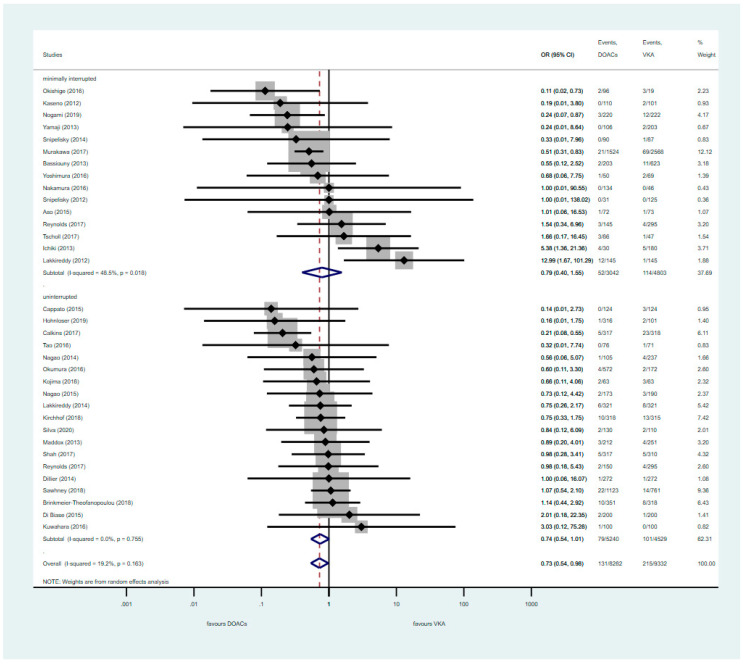
Composite of primary outcomes with uninterrupted or minimally interrupted DOAC strategies vs. VKA. CI: Confidence Interval; OR: Odds Ratio; DOAC: Direct Oral Anticoagulants; VKA: vitamin K antagonist. The diamond defines the overall result of each given group, the dotted red line is there to graphically show the overall results of the groups.

**Figure 5 jcm-09-03073-f005:**
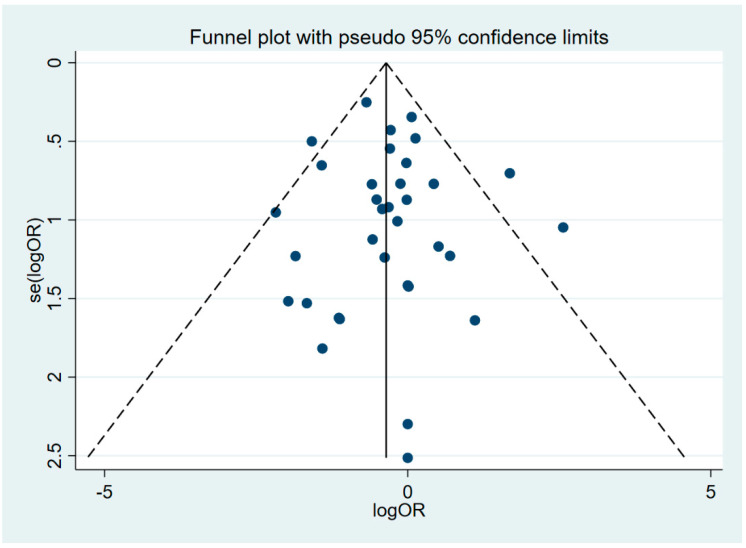
Funnel plot analysis. logOR, OR: Odds Ratio.

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
