# Peer review of "Uninterrupted or Minimally Interrupted Direct Oral Anticoagulant Therapy is a Safe Alternative to Vitamin K Antagonists in Patients Undergoing Catheter Ablation for Atrial Fibrillation: An Updated Meta-Analysis"

_jcm, 2020, doi:10.3390/jcm9103073_

Round 1

Reviewer 1 Report

The metaanalysis by Ottoffy et al. investigates different anticoagulation regimes in the context of CA for AF. The topic is of great importance for everyday clinical practice. By pooling data from 42 studies, a large group of patients was analysed with respect to safety and efficacy of peri-interventional anticoagulation.

However, there are some issues to be addressed in order to improve the manuscript, particularly in light of the recent guidelines and trials published during this year's ESC congress.

  1. Conducting CA for AF on uninterrupted OAK is the current guideline recommendation. Authors state, the clinical practice instead often involves minimal interruption or an interruption of NOAC therapy for 24 hours, probably because of subjectively perceived greater safety with respect to bleeding complications. Comparison of uninterrupted and interrupted NOAC therapy could be further highlighted in the main manuscript.
  2. Baseline characteristics of patients should be visualized more comprehensively and clearly, as well as  included in the main manuscript. Any statistically significant difference between subgroups should be accounted for in the analysis of study endpoints.
  3. Figure 1 could be simplified to avoid redundancies
  4. Introduction and Discussion should be revised incorporating recent guidelines and publications (e.g. EAST – AFNET)
  5. Overall, the Introduction could be shortened for a more concise overview.

Author Response

Answer to Reviewer 1:

  • Conducting CA for AF on uninterrupted OAK is the current guideline recommendation. Authors state, the clinical practice instead often involves minimal interruption or an interruption of NOAC therapy for 24 hours, probably because of subjectively perceived greater safety with respect to bleeding complications.
  1. Comparison of uninterrupted and interrupted NOAC therapy could be further highlighted in the main manuscript.

  1. Absolutely legitimate comment. According to the new 2020 ESC Guidelines for the diagnosis and management of atrial fibrillation, performance of the ablation procedure without OAC interruption is recommended (Class I A). We updated the introduction and discussion accordingly.
  2. Thanks for this suggestion. Previous consensus statement recommended uninterrupted or minimally interrupted dosing of DOAC during AF ablation. Most recent ESC guidelines clearly recommend performing the procedure without OAC interruption. In contrast, a survey in Europe (ESS-PRAFA) demonstrated that most centers apply a minimally interrupted DOAC dosing strategy to avoid bleeding complications due to an unpredictable effect of DOACs when no routine coagulation measurement is available. The majority of recent articles mainly focus on uninterrupted and minimally interrupted strategies. Interrupted studies were less recent and mainly retrospective, observational with lower level of evidence, so in our analysis we aimed to focus on interrupted and minimally interrupted therapeutical strategies, according to the usual daily practice. Nevertheless, we analyzed interrupted trials as well and discussed in chapter 3.3 and Fig. S7, in Supplementary Material.

  • Baseline characteristics of patients should be visualized more comprehensively and clearly, as well as included in the main manuscript. Any statistically significant difference between subgroups should be accounted for in the analysis of study endpoints.

  1. Due to the large amount of studies (and data) we were unable to assemble a solid and visually attractive/informative table into the main manuscript.
  2. We have now included the weighted average of the different characteristics of the subgroups in the main manuscript.
  3. No statistically significant differences were found in between the subgroups.

  • Figure 1 could be simplified to avoid redundancies

  1. We completely agree and excluded the superfluous steps in the figure.

  • Introduction and Discussion should be revised incorporating recent guidelines and publications (e.g. EAST – AFNET)
    1. Very important comment. The proof that AF ablation lowers major cardiovascular events was currently published (after original submission) in the EAST-AFNET 4 Trial. Early rhythm-control therapy was associated with a 21% lower risk of composite cardiovascular outcomes (death from cardiovascular causes, stroke, or hospitalization with worsening of heart failure or acute coronary syndrome) than usual care among patients with early atrial fibrillation and cardiovascular conditions. We updated the text and included the article in the corresponding sections.

  • Overall, the Introduction could be shortened for a more concise overview.

  1. We have updated the Introduction section with the most recent clinical evidence, and also shortened. Redundant/duplicate paragraphs, which we also discussed in the summary, have been removed. Discussion also shortened and restructured and updated accordingly. Redundant chapter about ACT variability was omitted. References was also updated. We have sent the manuscript for professional medical proofreading. We hope that the improved/updated text will meet your expectations.

Once again thank you for your kind and valuable suggestions. All these comments definitely improved the quality of the manuscript. We hope that our answers and corrections fulfill your requirements.

Sincerely yours,

Tamás Habon MD, PhD, FESC, FHFA and Máté Ottóffy MD

Reviewer 2 Report

It is an excellent well-balanced review. It is well illustrated. The manuscript is very well written and provides a comprehensive overview of the topic. In order for clarity to be the guiding principle, the authors should use active voice instead of passive voice. The sentences should be short and contain one point. The paragraphs should contain one theme, few sentences, and generally as few words as possible.

The topic is of interest and the results are of relevance.

Author Response

Answer to Reviewer 2:

  • It is an excellent well-balanced review. It is well illustrated. The manuscript is very well written and provides a comprehensive overview of the topic. In order for clarity to be the guiding principle, the authors should use active voice instead of passive voice. The sentences should be short and contain one point. The paragraphs should contain one theme, few sentences, and generally as few words as possible.

  1. Thank you for acknowledging our work and for your favorable feedback. We updated the text with the most recent clinical evidence, and we sent the manuscript for professional medical proofreading. We hope that the improved/updated text will meet your expectations.

Once again thank you for your kind and valuable suggestions. All these comments definitely improved the quality of the manuscript. We hope that our answers and corrections fulfill your requirements.

Sincerely yours,

Tamás Habon MD, PhD, FESC, FHFA and Máté Ottóffy MD

Round 2

Reviewer 1 Report

Authors replied adequately to the reviewers' remarks. Only minor comments:

p.4 ll. 137-139: add units of measurement

p. 9 l. 226: Romero et al. instead of "and his colleagues".

Author Response

Dear Reviewer!

Thank you for your perceptive remarks!

We have corrected the manuscript accordingly.

Sincerely yours,

On behalf of all authors,
Tamás Habon MD, PhD, FESC, FHFA and Máté Ottóffy MD